# Response of soil nutrient dynamics and stoichiometric characteristics in blueberry to fertilization rates

Xianyong Wang, Xiaoli An, Na Lei, Delu Wang[ID]*

College of Forestry, Guizhou University, Guiyang, Guizhou, China

* deluwang23@aliyun.com

## Abstract

To accurately understand the fertilizer requirements of blueberries during various growth stages, this study utilized 7-year-old rabbiteye blueberry cultivar 'Powderblue' as the research material. Based on leaf ecological stoichiometry, combined with photosynthetic rate and leaf area, the theoretical nitrogen (N), phosphorus (P), and potassium (K) requirements were calculated every 10 days across four growth stages: flowering (S1), fruit setting (S2), young fruit development (S3), and fruit expansion (S4). Fertilization experiments with 1× (1F), 5× (5F), 10× (10F), and 15× (15F) of the theoretical nutrient demands were conducted during these four stages to investigate the effects of varying NPK application rates on soil nutrient content throughout the annual growth cycle. Results indicated that high fertilizer treatments significantly increased short-term soil N, P, and K availability; however, their concentrations decreased by day 10 post-application, indicating the necessity for topdressing at 10-day intervals to maintain nutrient effectiveness. Soil organic carbon and organic matter levels tended to increase on day 5 post-fertilization but generally decreased by day 10, with more pronounced declines observed in low-rate treatments (e.g., 1F). Soil stoichiometric ratios remained relatively stable across fertilizer gradients, suggesting that plants absorbed nutrients proportionally while maintaining a balance of residual nutrients. Blueberries exhibited higher N uptake compared to P and K across all stages, with the fruit expansion stage demonstrating the highest nutrient absorption rates—2.95 to 10.55 times (N), 3.05 to 6.53 times (P), and 2.77 to 8.54 times (K) those of the preceding stages. These findings underscore the necessity of prioritizing nitrogen (N) supply while dynamically adjusting phosphorus (P) and potassium (K) application ratios during each growth phase, particularly during the fruit expansion stage. Furthermore, balancing soil organic matter mineralization and accumulation is crucial for achieving synergistic regulation of nutrient use efficiency and soil health.

**Data availability statement:** All relevant data are within the manuscript and its Supporting information files.

**Funding:** The research described in this manuscript was supported by Guizhou Provincial Science and Technology Plan Project (Qian Kehe Talents CXTD [2025] 053).

**Competing interests:** The authors have declared that no competing interests exist.

## 1. Introduction

Ecological stoichiometry is a discipline that focuses on the balance relationships among multiple chemical elements in ecosystems, playing a crucial role in the study of nutrient cycling among plants, microorganisms, and soil. It has been recognized as an important method for investigating nutrient cycling and dynamic changes in terrestrial ecosystems [1–3]. Soil carbon (C), nitrogen (N), phosphorus (P), and potassium (K) are essential chemical nutrients for plant growth and development, performing vital functions throughout the growth process [4,5]. Specifically, soil organic carbon (SOC) represents the largest carbon pool in terrestrial ecosystems and plays a critical role in global carbon cycling [6]. Soil nitrogen (N) serves as the primary component of proteins, which are closely associated with crop yield and quality while being the most dynamic element affecting soil fertility [7]. Soil phosphorus (P) is involved in the synthesis of energy substances such as adenosine triphosphate and reduced coenzymes in plants, indirectly facilitating the conversion of other elements and energy cycles [8]. Soil potassium (K) plays a crucial role in promoting photosynthesis, protein synthesis, and intracellular enzyme activation, thereby maintaining intracellular ion balance and regulating stomatal movement [9]. Within the soil system, stoichiometric relationships among carbon (C), nitrogen (N), phosphorus (P), and potassium (K) exhibit homeostasis, influencing soil microbial dynamics, plant root nutrient uptake, and nutrient cycling processes, which are essential for maintaining the structure, function, and stability of the entire soil system [10]. In a study examining the effects of nitrogen and phosphorus addition on soil nutrient content and the ecological stoichiometric characteristics of desert steppe, it was found that the desert steppe maintains a relatively stable C:N ratio, while the N:P ratio is primarily limited by soil phosphorus content [11]. Additionally, it was observed that as the age of Chinese fir forests increases, the soil C:N ratio remains unchanged, whereas the C:P and N:P ratios reach their maximum in 24-year-old mature forest [12]. Furthermore, altitude, slope, soil bulk density, soil water content and canopy density are identified as the main factors affecting the ecological stoichiometric characteristics of rubber forest soil [13]. Moreover, the contents and ratios of C, N, P, and K in the soil directly influence plant nutrient absorption and utilization, potentially altering biomass allocation and ecological strategies [14]. Therefore, studying the contents of soil C, N, P, and K and their ecological stoichiometric characteristics is critical for revealing the influencing factors and mechanisms underlying ecosystem structure, processes, and functions.

Blueberries (*Vaccinium* spp.), commonly referred to as bilberries or blueberries, belong to the Ericaceae family and are classified as perennial deciduous or evergreen shrubs [15]. These fruits are abundant in anthocyanins and offer numerous health benefits, including antihypertensive and lipid-lowering effects [16], anticancer properties [17], eye protection anti-inflammatory functions [18], and preventive effects against cardiovascular and neurodegenerative diseases [19,20]. As a result, they have been recognized by the United Nations Food and Agriculture Organization as one of the world's five healthiest fruits, earning the titles "Queen of Fruits" and "King of Berries" [21,22]. In China, however, irrational fertilization practices prevail

in blueberry cultivation, with growers relying excessively on empirical methods. Many mistakenly believe that increased fertilizer inputs directly correlate with higher yields, leading to over-fertilization, which can cause soil salinization, inhibit plant growth, and even result in mortality [23]. While blueberries require substantial nutrients during critical growth stages such as flowering and fruiting, their specific fertilizer demands at various developmental phases remain poorly understood. Therefore, this study conducts fertilization experiments grounded in ecological stoichiometry, aligned with the actual physiological requirements of blueberries, aiming to provide a theoretical foundation for scientific fertilization practices in blueberry production.

## 2. Materials and methods

### 2.1. Experimental materials

The experiment was conducted at the experimental base of Guizhou University, located in Guiyang City, Guizhou Province (106°27′-106°52′E, 26°11′-26°34′N). The test material consisted of 7-year-old rabbiteye blueberry (Vaccinium ashei Reade) cultivar 'Powderblue', with an average plant height of 84.26 cm, a crown width of 64.39 cm × 51.62 cm, and ground diameter of 12.09 mm. On December 23, 2022, the plants were transplanted into reinforced planting bags (top diameter 30 cm, bottom diameter 30 cm, height 25 cm), with one plant per bag. The cultivation substrate used was a low-nutrient acidic universal medium, composed of peat, quartz sand, and perlite in a volume ratio of 3:1:1. This substrate was characterized by total nitrogen (TN) of 0.58%, total phosphorus (TP) of 0.64 g/kg, total potassium (TK) of 4.61 g/kg, available nitrogen (AN) of 72.03 mg/kg, available phosphorus (AP) of 16.28 mg/kg, available potassium (AK) of 37.74 mg/kg, soil organic matter (SOM) of 501.97 g/kg, and a pH of 4.97.

### 2.2. Experimental design

Critical nutrient levels for rabbiteye blueberry (Vaccinium ashei) leaves have been established in the literature as follows: nitrogen (N) at 1.20% to 1.70%, phosphorus (P) at 0.08% to 0.17%, and potassium (K) at 0.28% to 0.60% [24,25]. Utilizing a leaf carbon (C) concentration of 45% as a reference [26,27], ecological stoichiometric ratios of rabbiteye blueberry leaves were calculated (see Table 1). The photosynthetic rate was measured using a Li-6800 portable photosynthetic apparatus (Beijing Ligaotai Technology Co., Ltd.) and the photosynthetic area was assessed with a YMJ series handheld leaf area meter (Shandong Holder Electronic Technology Co., Ltd.) across four developmental stages: flowering stage (S1), fruit setting stage (S2), young fruit stage (S3) and fruit expansion stage (S4) (refer to Table 2). Daily carbon production was calculated using the formula: total photosynthetic yield = photosynthetic rate × leaf area × photosynthetic duration [28]. Theoretical nutrient demands for each developmental stage were derived by integrating leaf stoichiometric ratios and subsequently converted to fertilizer requirements (see Table 3). The photosynthetic rate was averaged over 2–3 consecutive days, leaf area represented the average per plant, and the photosynthetic duration was standardized at 8 hours per day.

**Table 1. Presents the range of ecological stoichiometric ratios for rabbiteye blueberry leaves.**

| Stoichiometric ratio | Range | Mean value |
|---|---|---|
| C:N | 26.47~37.50 | 31.03 |
| C:P | 264.71~562.50 | 360.00 |
| C:K | 75.00~160.71 | 102.27 |
| N:P | 7.06~21.25 | 14.15 |
| N:K | 2.00~6.07 | 4.04 |
| K:P | 1.65~7.50 | 4.57 |

Note: The ranges of C:N, C:P and C:K in the table are derived from the "maximum and minimum values of carbon/ nutrient elements", while the average value is derived from the "carbon/ nutrition".

 

**Table 2. Presents the calculation parameters for fertilizer application at each growth stage.**

| Fertilizer time | Gross photosynthetic rate (µmol m$^{-2}$ s$^{-1}$) | Average leaf area per plant (cm$^2$) |
|---|---|---|
| S1-1 (2023.3.25) | 7.65 | 517.53 |
| S1-2 (2023.4.5) | 5.97 | 775.00 |
| S2-1 (2023.4.16) | 7.30 | 1325.82 |
| S3-1 (2023.4.26) | 9.27 | 1452.10 |
| S4-1 (2023.5.13) | 9.43 | 1481.20 |
| S4-2 (2023.5.23) | 10.21 | 1544.31 |
| S4-3 (2023.6.2) | 10.95 | 1485.73 |
| S4-4 (2023.6.12) | 11.81 | 1795.96 |

Note: Sn-m, where n represents the developmental periods 1–4, and m indicates the mth fertilization.

**Table 3. Details the amount of fertilizer applied to blueberries over a 10-day period during each stage (Compounds).**

| Fertilizer time | Varieties of fertilizer | Fertilization rate (g/ plant) | | | |
|---|---|---|---|---|---|
| | | 1F | 5F | 10F | 15F |
| S1-1 (2023.3.25) | N fertilizer | 0.2080 | 1.0401 | 2.0802 | 3.1204 |
| | P fertilizer | 0.0154 | 0.0772 | 0.1545 | 0.2317 |
| | K fertilizer | 0.0299 | 0.1494 | 0.2988 | 0.4482 |
| S1-2 (2023.4.5) | N fertilizer | 0.2431 | 1.2154 | 2.4308 | 3.6463 |
| | P fertilizer | 0.0180 | 0.0902 | 0.1805 | 0.2707 |
| | K fertilizer | 0.0349 | 0.1746 | 0.3492 | 0.5237 |
| S2-1 (2023.4.16) | N fertilizer | 0.5090 | 2.5452 | 5.0903 | 7.6355 |
| | P fertilizer | 0.0378 | 0.1890 | 0.3780 | 0.5669 |
| | K fertilizer | 0.0731 | 0.3656 | 0.7311 | 1.0967 |
| S3-1 (2023.4.26) | N fertilizer | 0.7075 | 3.5374 | 7.0749 | 10.6123 |
| | P fertilizer | 0.0525 | 0.2627 | 0.5253 | 0.7880 |
| | K fertilizer | 0.1016 | 0.5081 | 1.0162 | 1.5243 |
| S4-1 (2023.5.13) | N fertilizer | 0.7340 | 3.6701 | 7.3402 | 11.0103 |
| | P fertilizer | 0.0545 | 0.2725 | 0.5450 | 0.8175 |
| | K fertilizer | 0.1054 | 0.5272 | 1.0543 | 1.5815 |
| S4-2 (2023.5.23) | N fertilizer | 0.8289 | 4.1446 | 8.2891 | 12.4337 |
| | P fertilizer | 0.0615 | 0.3077 | 0.6155 | 0.9232 |
| | K fertilizer | 0.1191 | 0.5953 | 1.1906 | 1.7859 |
| S4-3 (2023.6.2) | N fertilizer | 0.8550 | 4.2748 | 8.5495 | 12.8243 |
| | P fertilizer | 0.0635 | 0.3174 | 0.6348 | 0.9522 |
| | K fertilizer | 0.1228 | 0.6140 | 1.2280 | 1.8420 |
| S4-4 (2023.6.12) | N fertilizer | 1.1147 | 5.5737 | 11.1474 | 16.7210 |
| | P fertilizer | 0.0828 | 0.4138 | 0.8277 | 1.2415 |
| | K fertilizer | 0.1601 | 0.8006 | 1.6012 | 2.4017 |

Note: Nitrogen fertilizer is represented by ammonium sulfate ($(NH_4)_2SO_4$, AR, ≥ 99%), phosphorus fertilizer by calcium superphosphate ($Ca(H_2PO_4)_2 \cdot H_2O$, AR, ≥ 92%), and potassium fertilizer by potassium sulfate ($K_2SO_4$, AR, ≥ 99%).

Due to the uncertainty surrounding the absorption and utilization efficiency of fertilizer by blueberry plants as well as the effective conversion efficiency of applied fertilizers in the soil, a discrepancy exists between the theoretical and actual fertilizer requirements. Consequently, it is essential to conduct gradient fertilization tests to verify these requirement. In

this study, four fertilizer gradients were established: 1× (1F), 5× (5F), 10× (10F), and 15× (15F) of the 10-day theoretical requirement, applied independently across four developmental stages (with no cross-treatment). This resulted in a total of 16 treatments (4 stages × 4 gradients), each replicated three times (7 plants per replicate), culminating in 336 plants. Fertilizers (chemical compounds) were dissolved in 700 mL of water and applied via slow irrigation. The test plants were randomly placed in a plastic film greenhouse for cultivation. The arrangement of plants for the four treatments was organized in a 12 × 7 layout, consisting of 12 columns with 7 plants in each column. With the exception of the experimental treatment, all other management measures were standardized. The definitions of the developmental stages are as follows:

Flowering (S1): 5% to 90% flower bloom;

Fruit setting (S2): 5% to 90% fruit set;

Young fruit (S3): Post-90% fruit set to the initiation of rapid expansion in 5% of the fruit;

Fruit expansion (S4): From the initiation of rapid expansion in 5% of the fruit to 5% fruit maturity.

## 2.3. Sample collection and determination

Soil samples were collected the day before each fertilization and on the 5th and 10th days following each fertilization. A soil sampler (total length of 35 cm, diameter of 10 mm) was used to collect samples, with four points taken from each pot of seedlings (see Fig 1). Three biological replicates were obtained for each treatment, with 100 g of soil collected for each replicate.

Soil organic carbon (SOC) and organic matter (SOM) contents were determined using the oil bath-heated potassium dichromate oxidation method. Total nitrogen (TN) was measured through sulfuric acid-perchloric acid digestion, followed by alkaline hydrolysis diffusion and semi-micro Kjeldahl analysis. Total phosphorus (TP) was quantified using the molybdenum-antimony colorimetric method, while total potassium (TK) was analyzed via flame photometry. Available nitrogen (AN) was determined through alkaline hydrolysis diffusion combined with semi-micro Kjeldahl analysis. Available phosphorus (AP) was measured using sodium bicarbonate extraction and molybdenum-antimony colorimetry, and available potassium (AK) was analyzed via ammonium acetate extraction and flame photometry. Nutrient uptake calculations were performed as follows: Nutrient uptake per plant during a stage = Initial soil nutrient content at the stage's start + Fertilizer nutrient input during the stage – Final soil nutrient content at the stage's end; Daily nutrient uptake rate = Total nutrient uptake per plant/ Number of days in the stage.

## 2.4. Data processing

Data sorting, calculations, and mapping were conducted using Excel 2019 and Origin 2024. One-way analysis of variance (ANOVA) was performed on the data utilizing SPSS version 19.0. The Least Significant Difference (LSD) method was employed to compare differences among various data groups. A significance level of $\alpha = 0.05$ was established, and statistical differences were indicated using the sequential letter labeling method.

## 3. Results and analysis

### 3.1. Effect of fertilizer application rate on soil nitrogen content

As demonstrated in Fig 2, independent fertilization at each growth stage significantly influenced the contents of total nitrogen (TN) and alkaline hydrolyzable nitrogen (AN) in blueberry plants. Following each fertilization during the flowering stage, both TN and AN contents exhibited an increase on day 5, followed by a decrease on day 10 across all fertilizer gradients. Specifically, on day 5 after the initial fertilization, TN contents rose by 10.36% to 25.20% and AN by 7.37% to 113.99% compared to pre-fertilization levels, with the 15F treatment yielding the hightest increases. After the second

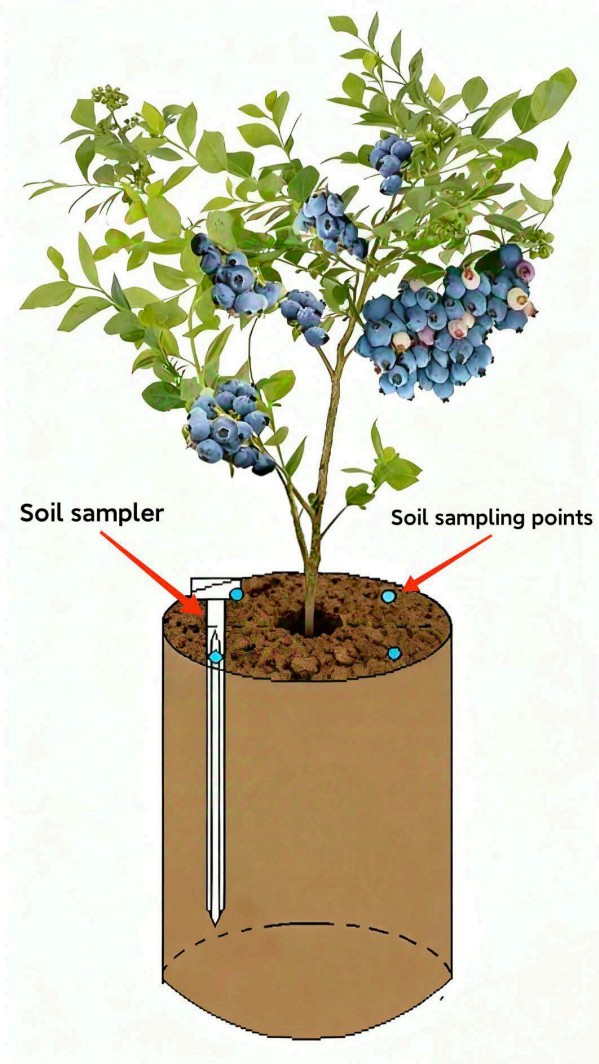

**Fig 1. Schematic diagram of soil sample collection.**

fertilization, TN increased by 9.45% to 19.23% and AN by 8.36% to 27.13% on day 5, with the most significant TN increase observed in the 15F treatment and the largest AN increase noted in the 10F treatment.

During the fruit setting stage, total nitrogen (TN) contents increased by 12.99%, 24.15%, 30.93%, and 30.27% in the 1F to 15F treatments respectively on day 5 post-fertilization. This was followed by decreases of 8.34%, 17.21%, 22.16%, and 21.42% on day 10. Similarly, amino nitrogen (AN) contents rose by 18.27%, 57.63%, 151.02%, and 202.57% on day 5, then declined by 8.3%, 22.50%, 36.24%, and 26.37% on day 10. After fertilization during the young fruit stage, TN contents increased by 3.74% to 34.22%, while AN increased by 4.43% to 361.28% on day 5. Although both nutrients decreased from day 5 to day 10, most post-decline values remained higher than pre-fertilization levels, with the exception of TN and AN in the 1F treatment, and TN in the 5F treatment. During the fruit expansion stage, TN and AN contents exhibited a consistent pattern of increases on day 5 and decreases on day 10 across all four fertilization treatments. Fig 1 also illustrates cumulative increases in TN and AN from May 18 to June 17, with higher increments observed in treatments

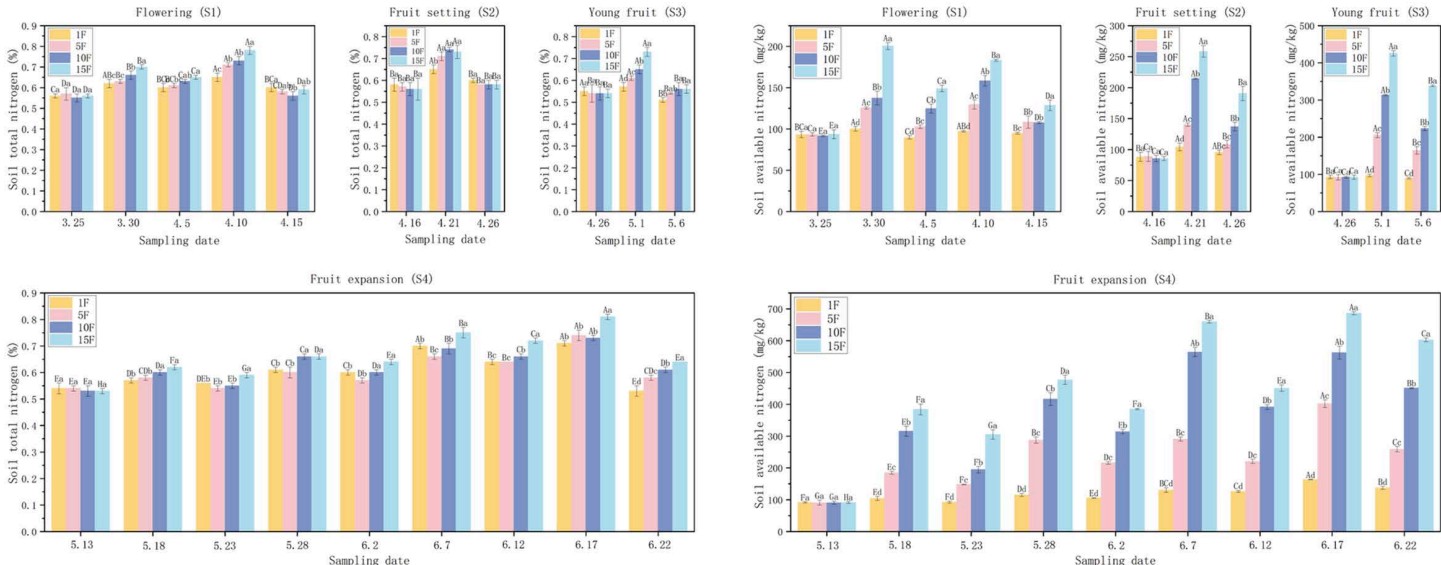

**Fig 2. Effects of different fertilization rates on soil TN and AN contents in different periods.** Note: Different uppercase letters in the figure indicate that the differences between the same treatment at different times are significant, while different lowercase letters denote that the differences between different treatments at the same time are significant ($P<0.05$).

receiving more frequent fertilization. Compared to pre-fertilization baselines, day 5 TN increases ranged from 6.53% to 16.44%, 9.42% to 18.96%, 14.31% to 18.25%, and 10.53% to 15.17% across fertilizer gradients, while AN increases spanned from 14.41% to 321.51%, 24.82% to 114.02%, 23.88% to 79.75%, and 29.49% to 82.04%. Notably, AN demonstrated greater proportional increases relative to TN across all treatments.

### 3.2. Effect of fertilizer application rate on soil phosphorus content

As shown in Fig 3, during the flowering stage, the total phosphorus (TP) content in the soil remained relatively stable, ranging from 0.51 to 0.57 g/kg, irrespective of the timing of fertilization. The available phosphorus (AP) exhibited increases of 0.04% to 46.61% and 2.69% to 31.97% on day 5 following the first and second fertilizations respectively; however, it decreased by day 10 while still generally exceeding pre-fertilization levels. Notable exceptions included the AP levels in the 1F and 5F treatments on April 5, as well as the 1F treatment on April 15, indicating an insufficient phosphorus supply to meet the requirements over the 10-day period.

During the fruit-setting stage, TP increased by 2.57% to 8.16% on day 5 post-fertilization. By day 10, the TP levels in the 1F, 5F, and 10F treatments fell below pre-fertilization levels, while the 15F treatment maintained a higher TP content. The AP increased by 5.13% to 93.01% on day 5 (with the maximum observed in the 15F treatment), but decreased by day 10 with only the 1F treatment falling below baseline levels. Following fertilization during the young fruit stage, TP increased by 3.40% to 10.55% on day 5. By day 10, TP levels in the 5F, 10F, and 15F treatments remained 2.05%, 2.26%, and 3.48% above pre-fertilization levels, while the 1F treatment decreased by 1.30%. The AP increased rapidly by 5.94%, 24.01%, 61.03%, and 82.46% across the 1F to 15F treatments on day 5, before declining by 10.00% to 14.37% on day 10. Only the AP in the 1F treatment fell 6.01% below baseline, while the other treatments remained elevated. During the fruit expansion stage, both TP and AP contents exhibited significant treatment×time interactions. Across the four fertilization treatments, both nutrients demonstrated a consistent pattern similar to nitrogen: increases on day 5 followed by decreases on day 10, highlighting the necessity of phosphorus topdressing at 10-day intervals to maintain nutrient

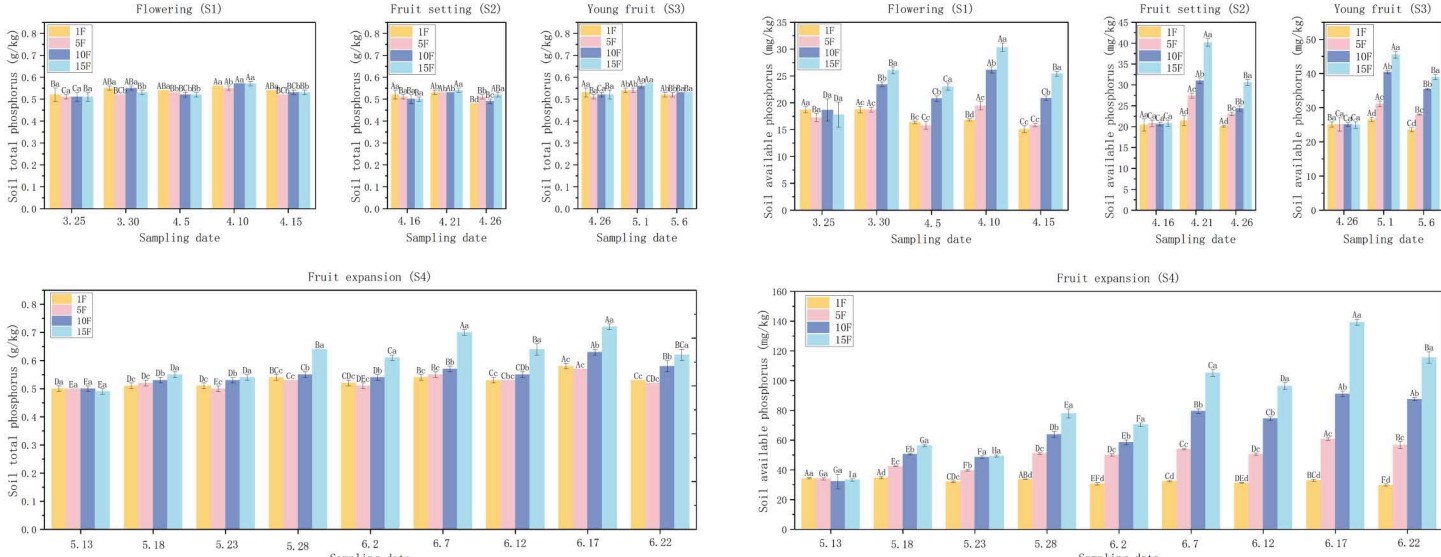

**Fig 3. Effects of different fertilization rates on soil TP and AP contents in different periods.** Note: Different uppercase letters in the figure indicate that the differences between the same treatment at different times are significant, while different lowercase letters denote that the differences between different treatments at the same time are significant ($P<0.05$).

availability. In contrast to nitrogen, the increases in TP and AP on day 5 (ranging from 2.46% to 17.85% and 0.94% to 69.12%, respectively) were smaller in magnitude compared to total nitrogen (TN) and ammonium nitrogen (AN).

### 3.3. Effect of fertilizer application rate on soil potassium content

As illustrated in Fig 4, the application of varying amounts of nitrogen, phosphorus, and potassium fertilizers resulted in significant differences in both total potassium and available potassium content in the soil, attributable to the differing fertilization rates. Furthermore, the trends observed in total potassium and available potassium content following the four fertilization treatments were consistent with those of nitrogen and phosphorus.

### 3.4. Effect of fertilization on soil organic carbon and organic matter content

As illustrated in Fig 5, soil organic carbon (SOC) and organic matter (SOM) demonstrated consistent trends across various fertilization treatments throughout each growth stage. During the flowering stage, SOC and SOM increased in all treatments on day 5 following the initial fertilization. By day 10, treatments 10F and 15F continued to exhibit increases, while treatments 1F and 5F experienced a decline from their day 5 levels. Following the second fertilization, both SOC and SOM decreased in the 1F, 10F, and 15F treatments on days 5 and 10, whereas treatment 5F showed an increase on day 5, followed by a decrease on day 10.

During the fruit setting stage, SOC increased by 3.21% and 1.42% in the 5F and 15F treatments on day 5, respectively, but decreased by 1.80% and 8.31% in the 1F and 10F treatments. By day 10, all treatments exhibited SOC levels below pre-fertilization levels. SOM mirrored the trends observed in SOC on both days 5 and 10. After fertilization during the young fruit stage, SOC and SOM declined in the 1F, 5F, and 15F treatments on both days 5 and 10, with treatment 5F showing a decrease on day 5 followed by an increase on day 10. By day 10, SOC and SOM had dropped by 10.55% to 14.15% compared to pre-fertilization levels. During the fruit expansion stage, following four fertilizations, SOC and SOM decreased on day 5 after the first two fertilizations, increased after the last two fertilizations, and then decreased again by day 10. All treatments reached their minimum SOC and SOM levels on June 12, which was day 10 after the third fertilization.

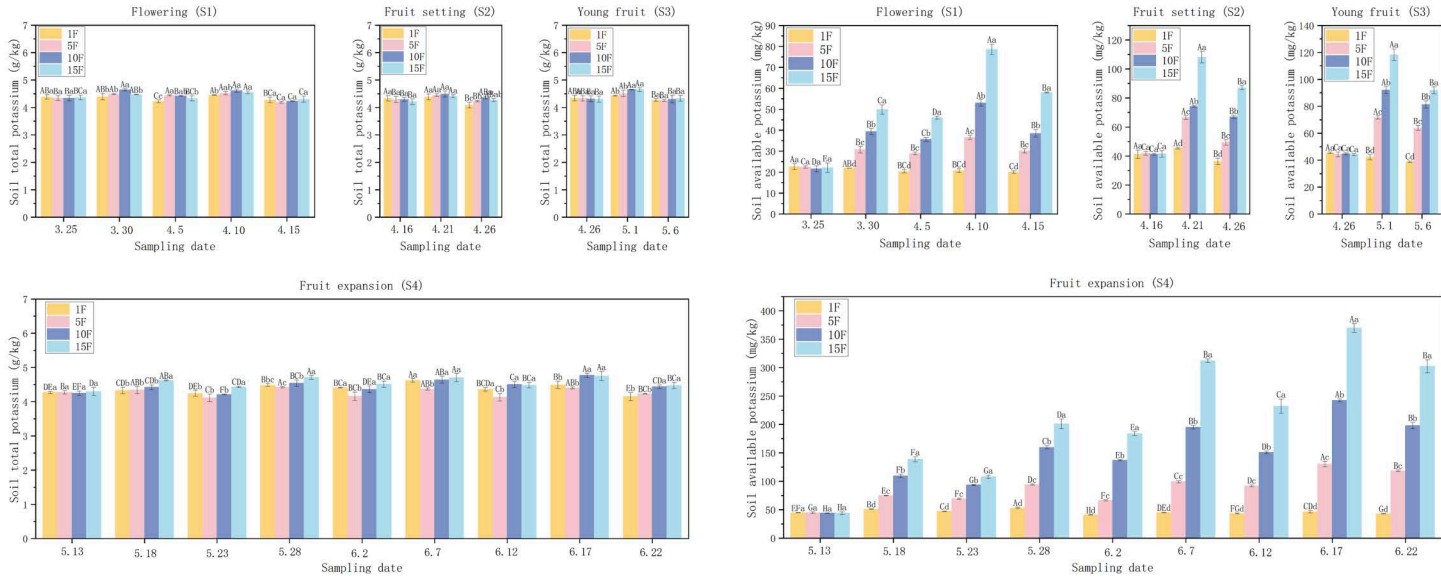

**Fig 4. Effects of different fertilization rates on soil TK and AK contents in different periods.** Note: Different uppercase letters in the figure indicate that the differences between the same treatment at different times are significant, while different lowercase letters denote that the differences between different treatments at the same time are significant (*P* < 0.05).

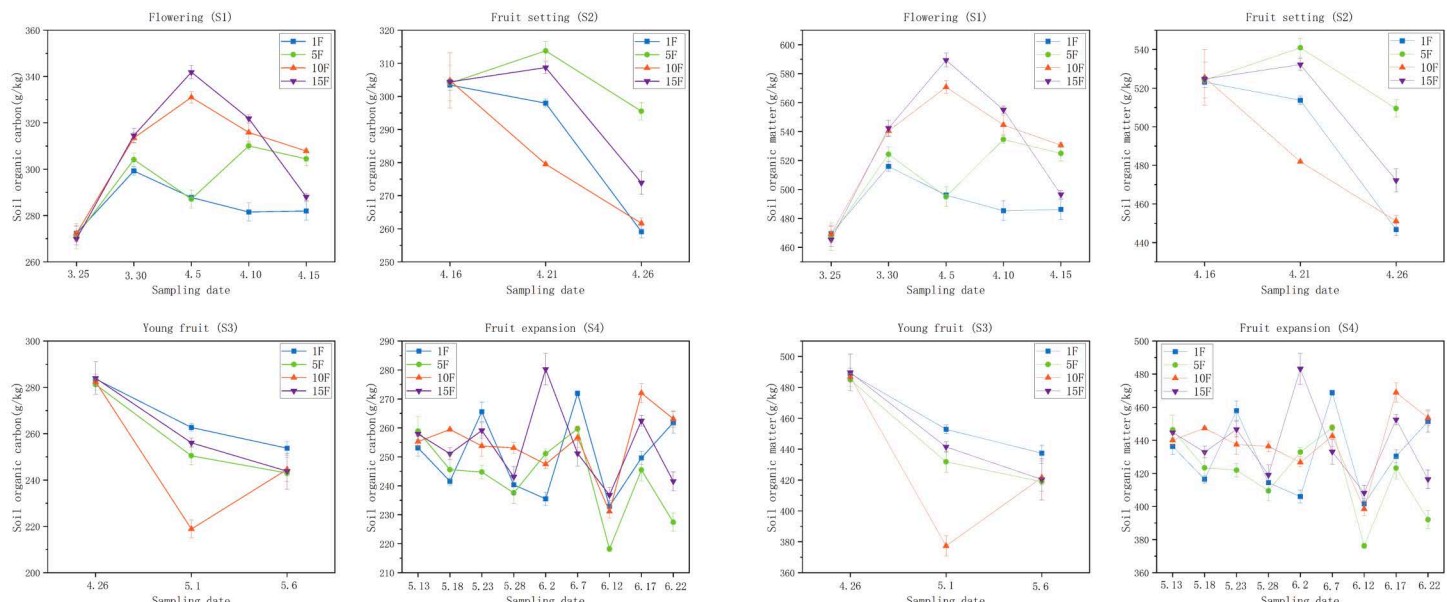

**Fig 5. Effects of different fertilization rates on soil SOC and SOM contents in different periods.** Note: Different uppercase letters in the figure indicate that the differences between the same treatment at different times are significant, while different lowercase letters denote that the differences between different treatments at the same time are significant (*P* < 0.05).

### 3.5. Effect of fertilizer application rate on soil stoichiometric ratio

As shown in Table 4, independent fertilization during each growth stage exerted significant effects on soil stoichiometric ratios. However, numerical fluctuations among treatments within the same stage remained minimal, indicating that stoichiometry-based fertilization maintained nutrient proportions within stable ranges. During the flowering stage, the soil C:N, C:P, and C:K ratios in the 5F and 10F treatments were significantly higher than those in other gradients. The N:P (10.63 to 11.30), N:K (1.33 to 1.40), and K:P (7.82 to 8.17) ratios showed no significant differences across treatments. After fruit setting fertilization, the 5F treatment achieved the highest C:N (50.59), C:P (583.49), and C:K (69.81) ratios, significantly exceeding those of other treatments. The maximum N:P, N:K, and K:P ratios were observed in the 1F, 1F, and 10F treatments respectively, closely related to the pronounced SOC enhancement in the 5F treatment. During the young fruit and fruit expansion stages, the 1F treatment exhibited the highest C:N, C:P, and C:K ratios. This was a result of reduced fertilizer inputs that slowed SOC mineralization and decomposition, combined with lower residual N/P/K after plant uptake. Across these stages, the N:P (9.90 to 10.51; 9.97 to 11.10), N:K (1.21 to 1.29; 1.29 to 1.44), and K:P (7.9 to 8.21; 7.16 to 8.08) ratios remained within reasonable ranges, indicating stable stoichiometry despite varying fertilizer rates. This suggests that plants absorbed nutrients proportionally, maintaining consistent residual nutrient ratios.

### 3.6. Effect of fertilization on the absorption of mineral elements in blueberry

As illustrated in Table 5, significant differences in the uptake of mineral nutrients (N, P, K) by blueberries were observed across various developmental stages and fertilizer treatments. Nutrient uptake exhibited a proportional increase with fertilizer application rates, indicating a dose-dependent relationship. Blueberries demonstrated stage-specific nutrient requirements, reflecting both temporal variations for the same element and inter-element

**Table 4. Effects of different fertilizer application rates on soil stoichiometric ratio characteristics at different stages.**

| Treatment | | C:N | C:P | C:K | N:P | N:K | K:P |
|---|---|---|---|---|---|---|---|
| Fertilizing season | Fertilizer amount | | | | | | |
| S1 | 1F | 47.33±0.56c | 519.46±4.37c | 66.04±0.60b | 10.98±0.22ab | 1.40±0.01a | 7.82±0.05c |
| | 5F | 52.86±0.52b | 584.57±3.59a | 72.92±0.53a | 11.06±0.15a | 1.38±0.01a | 8.02±0.05b |
| | 10F | 54.94±1.38a | 583.86±6.18a | 72.77±0.05a | 10.63±0.23b | 1.33±0.03b | 8.02±0.09ab |
| | 15F | 48.89±1.48c | 547.00±3.91b | 66.94±1.34b | 11.30±0.15a | 1.37±0.02a | 8.17±0.12a |
| S2 | 1F | 43.48±0.66d | 538.82±2.56b | 63.41±1.09b | 12.39±0.23a | 1.46±0.01a | 8.50±0.18b |
| | 5F | 50.59±0.14a | 583.49±3.47a | 69.81±0.19a | 11.53±0.07b | 1.38±0.01b | 8.36±0.05bc |
| | 10F | 45.49±1.39c | 530.85±4.23c | 59.95±0.33c | 11.78±0.10b | 1.32±0.03c | 8.85±0.05a |
| | 15F | 47.63±0.78b | 522.80±1.65d | 64.09±0.56b | 10.98±0.17c | 1.35±0.03bc | 8.16±0.10c |
| S3 | 1F | 49.35±0.59a | 488.44±1.27a | 59.48±0.08a | 9.90±0.12b | 1.21±0.01b | 8.21±0.02a |
| | 5F | 45.02±0.08b | 465.37±3.60b | 57.18±0.26b | 10.34±0.07a | 1.27±0.01a | 8.14±0.06ab |
| | 10F | 43.94±1.10bc | 459.23±7.76b | 56.86±1.05b | 10.31±0.25a | 1.29±0.03a | 7.99±0.07c |
| | 15F | 43.53±0.50c | 457.30±12.32b | 56.36±0.39b | 10.51±0.27a | 1.29±0.02a | 8.05±0.09bc |
| S4 | 1F | 49.15±1.51a | 496.14±4.61a | 63.17±0.95a | 9.97±0.16d | 1.29±0.03c | 7.86±0.19ab |
| | 5F | 39.17±0.36c | 434.89±2.55c | 53.81±0.59c | 11.10±0.06a | 1.37±0.02b | 8.08±0.05a |
| | 10F | 43.15±0.56b | 455.01±12.99b | 59.25±0.10b | 10.62±0.08b | 1.37±0.02b | 7.68±0.22b |
| | 15F | 37.56±0.26d | 387.21±5.52d | 54.06±0.46c | 10.31±0.20c | 1.44±0.02a | 7.16±0.05c |

Note: The stoichiometric ratio presented in the table reflects the relationship between soil organic carbon and total nutrient content. The soil nutrient data for S1 through S4 were collected on April 15, 2023, April 26, 2023, May 6, 2023, and June 22, 2023, respectively, marking the conclusion of the four developmental periods. The distinct lowercase letters within the same column of the table signify significant differences among various fertilization treatments during the same period ($P < 0.05$). The values reported are expressed as the mean±standard deviation.

**Table 5. Effects of different fertilizer application rates at different stages on mineral element absorption and daily absorption of blueberry.**

| Treatment | | N uptake(mg) | P uptake(mg) | K uptake(mg) | N daily uptake(mg) | P daily uptake(mg) | K daily uptake(mg) |
|---|---|---|---|---|---|---|---|
| Fertilizing season | Fertilizer amount | | | | | | |
| S1 | 1F | 77.16±0.52d | 30.08±2.37d | 47.33±0.61d | 3.67±0.02d | 1.43±0.11d | 2.25±0.03d |
| | 5F | 388.61±33.31c | 49.38±2.69c | 98.56±3.00c | 18.51±1.59c | 2.35±0.13c | 4.69±0.14c |
| | 10F | 859.07±6.22b | 64.52±2.99b | 188.64±4.61b | 40.91±0.30b | 3.07±0.14b | 8.98±0.22b |
| | 15F | 1224.41±29.08a | 72.56±2.45a | 219.80±12.40a | 58.31±1.38a | 3.46±0.12a | 10.47±0.59a |
| S2 | 1F | 79.14±2.06d | 14.41±2.32d | 62.19±5.30d | 7.91±0.21d | 1.44±0.23d | 6.22±0.53d |
| | 5F | 421.00±13.97c | 33.02±2.59c | 118.57±3.15c | 42.10±1.40c | 3.30±0.26c | 11.86±0.32c |
| | 10F | 770.61±21.47b | 70.68±1.59b | 172.14±3.98b | 77.06±2.15b | 7.07±0.16b | 17.21±0.40b |
| | 15F | 995.42±78.31a | 80.88±1.50a | 217.51±5.40a | 99.54±7.83a | 8.09±0.15a | 21.75±0.54a |
| S3 | 1F | 160.03±8.78d | 21.97±1.41d | 85.60±1.14d | 16.00±0.88d | 2.20±0.14d | 8.56±0.11d |
| | 5F | 314.17±22.66c | 51.06±6.00c | 109.90±4.46c | 31.42±2.27c | 5.11±0.60c | 10.99±0.45c |
| | 10F | 714.49±27.54b | 67.38±2.00b | 235.12±11.40b | 71.45±2.75b | 6.74±0.20b | 23.51±1.14b |
| | 15F | 773.73±22.02a | 109.69±2.69a | 395.89±10.83a | 77.37±2.20a | 10.97±0.27a | 39.59±1.08a |
| S4 | 1F | 471.40±17.00d | 91.72±0.54d | 237.11±2.39d | 11.78±0.42d | 2.29±0.01d | 5.93±0.06d |
| | 5F | 2736.22±11.58c | 186.67±9.88c | 694.62±2.28c | 68.41±0.29c | 4.67±0.25c | 17.37±0.06c |
| | 10F | 5323.94±20.04b | 312.89±23.17b | 1348.67±32.43b | 133.10±0.50b | 7.82±0.58b | 33.72±0.81b |
| | 15F | 8159.06±20.92a | 474.03±18.39a | 1857.65±56.90a | 203.98±0.52a | 11.85±0.46a | 46.44±1.42a |

Note: The absorption of mineral elements presented in the table is calculated based on the content of available nutrients. The different lowercase letters within the same column signify significant differences among various fertilization treatments during the same period ($P<0.05$), with the values representing the mean±standard deviation.

differences within the same stage. Notably, nitrogen (N) requirements consistently surpassed those of phosphorus (P) and potassium (K), with P exhibiting the lowest demand. Nutrient requirements progressively increased throughout development, peaking during the fruit expansion phase when N, P, and K uptake reached 2.95 to 10.55 times, 3.05 to 6.53 times, and 2.77 to 8.54 times those of the preceding stages respectively. Daily absorption rates during fruit expansion were 0.74 to 3.70 times (N), 0.91 to 3.43 times (P), and 0.69 to 4.44 times (K) higher than those of earlier stages.

## 4. Discussion

### 4.1. Response of soil nutrient content and stoichiometric ratio to fertilization amount

Fertilization increases exogenous nutrient inputs, enhances soil structure and fertility, and accelerates nutrient cycling [29]. In this study, following independent fertilization at four developmental stages, the total and available nitrogen (N), phosphorus (P), and potassium (K) contents in the 5F to 15F treatments were significantly higher than those observed prior to fertilization. This indicates that fertilization during the four developmental stages of blueberry effectively improves soil nutrient content. These findings align with previous studies [30–33]. During the fruit expansion phase, the availability of N, P, and K followed the trend 15F > 10F > 5F > 1F, suggesting dose-dependent relationships. Higher fertilizer rates provided more labile nutrients for fruit development, corroborating Liu Shuangfeng's report [33] that greater available P increases occur with higher P application rates. Although soil organic carbon (SOC) and soil organic matter (SOM) fluctuated with varying fertilizer rates and application frequencies during fruit expansion, the 1F to 15F treatments maintained relatively high levels. This suggests that a stoichiometry-based fertilization approach with a 10-day interval replenishes nutrients absorbed by plants and supports SOC/SOM retention, consistent with prior studies [34–36].

Soil stoichiometric ratios serve as indicators of nutrient status, fertility, and organic matter decomposition. The C:N, C:P, and C:K ratios provide insights into mineralization and immobilization processes, where lower ratios signify enhanced mineralization and nutrient release [37]. The reported average stoichiometric ratios for Chinese soils are 11.9 for C:N, 61 for C:P, and 5.2 for N:P [38]. Prescott [39] identified that N mineralization diminishes when the C:N ratio in forest soils exceeds 35. Additionally, Jia Yu [40] noted that net organic phosphorus mineralization occurs at C:P ratios below 200, while phosphorus immobilization is observed at C:P ratios above 300. In our study, post-fertilization ratios of C:N (>35), C:P (>300), and N:P (>5.2) exhibited significant variation, likely attributable to substrate texture and/or insufficient available nutrients. However, it is important to note that this study primarily focused on the stoichiometric ratios of blueberry leaves under different fertilization treatments. The soil stoichiometric characteristics, including C:N, C:P, C:K, N:P, N:K, and K:P ratios, showed minimal fluctuations across the study periods. This stability suggests that the nitrogen, phosphorus, and potassium ratios can enhance the soil's capacity to retain fertilizers and supply nutrients, potentially mitigating the risk of blueberry plants experiencing single salt toxicity.

### 4.2. Effect of fertilization on the absorption of mineral elements in blueberry

The uptake of plant nutrients from the soil is a continuous process throughout growth, influenced by fertilizer application rates, stage-specific demands, and complex synergistic and antagonistic interactions among nutrients, as well as between nutrients and soil [41]. Sun Mei [42] investigated nutrient uptake in Muscat Hamburg grapes under varying nutrient solution concentrations, revealing that the absorption of nitrogen (N), phosphorus (P), and potassium (K) followed a dose-dependent pattern of 1.5×>1×>0.5× across the flowering, young fruit, expansion, coloring, and ripening stages. This finding aligns with our results, which indicate increased N, P, and K uptake, along with daily absorption rates in blueberries, corresponding to higher fertilizer inputs during all four growth stages. Throughout these stages, blueberries consistently exhibited the highest N uptake, corroborating previous reports [43] that highlight the critical nitrogen demands during development. The pronounced peak in nutrient absorption during the fruit expansion phase corresponds to the rapid growth of the fruit, which necessitates substantial nutrient accumulation. Under the 15F treatment, maximum uptake values of 8.16 g N, 0.47 g P, and 1.86 g K were recorded, translating to 38.50 g of ammonium sulfate, 1.93 g of superphosphate, and 4.15 g of potassium sulfate. This underscores the necessity for timely NPK supplementation during the fruit expansion phase. Notably, the N, P, and K uptake in the 5F to 15F treatments generally lagged behind fertilizer inputs, suggesting either low nutrient use efficiency or limited bioavailability of soil nutrients. Future studies could investigate the potential of organic fertilizer amendments [44] or the inoculation of dark septate endophytes (DSE) [45] to enhance nutrient absorption efficiency.

### 5. Conclusion

The application of varying fertilizer rates based on leaf stoichiometric ratios during the four developmental stages of blueberry cultivation effectively increased the soil's available nitrogen, phosphorus, and potassium contents while maintaining stoichiometric stability. The available nutrient contents exhibited dose-dependent relationships with fertilizer inputs across all four stages. Blueberry nutrient uptake increased proportionally with fertilizer rates, demonstrating a clear dose-dependent relationship for mineral nutrient absorption. Notably, nitrogen uptake consistently surpassed that of phosphorus and potassium throughout all stages, with the highest absorption of nitrogen, phosphorus, and potassium occurring during the fruit expansion stage. These findings underscore the significance of stage-specific fertilization management, emphasizing the prioritization of nitrogen supply while ensuring timely supplementation of phosphorus and potassium. Furthermore, balancing soil organic matter accumulation and mineralization is critical for maintaining nutrient availability and promoting soil health.

### Supporting information

**S1 File. Dataset (Figs 2–5).**
(XLSX)

## Author contributions

**Data curation:** Xianyong Wang, Xiaoli An, Na Lei.

**Formal analysis:** Xianyong Wang, Xiaoli An, Na Lei.

**Methodology:** Delu Wang.

**Project administration:** Delu Wang.

**Resources:** Delu Wang.

**Software:** Xianyong Wang.

**Supervision:** Delu Wang.

**Validation:** Delu Wang, Xianyong Wang.

**Visualization:** Xianyong Wang.

**Writing – original draft:** Xianyong Wang, Xiaoli An.

**Writing – review & editing:** Delu Wang.

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
