## [Decision Letter · Decision Letter 0]

1 Aug 2025

Dear Dr. Wang,

Thank you for submitting your manuscript to PLOS ONE. After careful consideration, we feel that it has merit but does not fully meet PLOS ONE’s publication criteria as it currently stands. Therefore, we invite you to submit a revised version of the manuscript that addresses the points raised during the review process.

We look forward to receiving your revised manuscript.

Kind regards,

Rachid Bouharroud

Academic Editor

PLOS ONE

Journal Requirements:

2. Please include captions for your Supporting Information files at the end of your manuscript, and update any in-text citations to match accordingly. Please see our Supporting Information guidelines for more information: http://journals.plos.org/plosone/s/supporting-information .

Reviewers' comments:

Reviewer's Responses to Questions

**Comments to the Author**

1. Is the manuscript technically sound, and do the data support the conclusions?

Reviewer #1: Yes

Reviewer #2: Yes

2. Has the statistical analysis been performed appropriately and rigorously?

Reviewer #1: Yes

Reviewer #2: Yes

3. Have the authors made all data underlying the findings in their manuscript fully available?

Reviewer #1: Yes

Reviewer #2: Yes

4. Is the manuscript presented in an intelligible fashion and written in standard English?

Reviewer #1: No

Reviewer #2: Yes

Reviewer #1: In the Materials and Methods section, please specify the instruments and tools used to measure photosynthetic rate and leaf area. Details such as brand, model, and measurement protocols are important for reproducibility.

In the Experimental Design subsection, although you provided the critical nutrient concentrations in leaves, you did not clearly explain how these values were used to calculate the theoretical nutrient requirements of the plants. Additionally, please justify why nutrient concentrations in leaves alone were considered, rather than including other plant organs (e.g., fruits, roots, stems). A whole-plant nutrient accounting approach would provide a more comprehensive estimation of actual requirements.

Furthermore, clarify how the fertilizer application rates (in Table 3) were calculated from the theoretical NPK nutrient demands. The conversion from nutrient requirement to actual fertilizer quantity should be detailed, including molecular weights, purity, and fertilizer formulations used.

Check for missing or improperly expressed units throughout the text and in all tables. Consistency and clarity in unit expression are essential for interpreting results correctly.

Clarify the fertilizer treatment levels. Does "1F" represent the exact theoretical nutrient demand (i.e., the minimum requirement for optimal mineral nutrition of the crop)? Please explain the rationale behind the 1F, 5F, 10F, and 15F treatments.

In the Discussion on nutrient uptake, carefully re-evaluate the conclusion that nitrogen (N) is the most important element during the fruit expansion stage. In many fruit crops, nitrogen primarily supports vegetative growth and vigor, while potassium (K) is more closely associated with fruit size, firmness, and flavor, and phosphorus (P) with root development and flowering. Consider whether your findings for blueberry differ from this general pattern, and if so, explain the physiological basis.

Strengthen the Discussion section further by adding more context, critical analysis, and possible future research perspectives.

Please correct the many syntax, grammar, and vocabulary errors present in the manuscript. The current language quality significantly affects the clarity and readability of the paper and should be revised by a fluent English speaker or professional editor.

Key Revisions Suggested :

Clearly define the research hypothesis and scientific objectives in the introduction.

Perform a thorough English language revision to improve clarity, grammar, and scientific tone.

Strengthen the discussion section with critical interpretation and broader implications.

Reviewer #2: The description of the research is more than adequate and detailed. Regarding sustainable food production, food security, dangers from over fertilization, economical (inc. yield) aspects and the fact that in many regions irrational fertilization practices are deployed, this paper serves great impact on contributing to creating an efficient fertilization strategy.

The paper is structured well, following IMRaD structure. Introduction explains the background: importance of the different nutrients, their role, ratios. The authors have brought out the aspects of geographical conditions and a brief overview on importance of blueberries in general, including the delicate matters of their fertilization especially regarding outcomes of potential over fertilization. In materials and methods, the whole procedure is described: starting with location, plant type/cultivar, plant size, plant amount (336!!), timeframe, plantation conditions, followed by the before-during-after measurement setup and the description of each test, data collection and processing. Everything is described in a way that the experiment could be repeated in other geographical location with other cultivars. The results are brought out in a structured and detailed form, focusing on the NPK, SOC and SOM in soil, also NPK absorption in plant. Based on the results, discussion is divided into two aspects: stoichiometric fertilization and the absorption of mineral elements in the plants. The discussion can be interpreted into various suggestions how to fertilize or which criteria to consider, but does not offer a distinctive fertilization strategy. In general, the research reflects major practical work, comprehensive data collection, processing, interpreting and based on that, the paper offers knowledge for agronomists, farmers and even agricultural machinery engineers who could potentially develop machinery for precision fertilization.

1) As the it was described, the 7 year old plants of "Powderblue" cultivar are up to 85 cm tall. If and how could be the research results interpreted in order to apply NPK for smaller cultivars, such as Vaccinium angustifolium?

2) Which was the plantation pattern or the layout? What was the distance from the plants in row and column? Perhaps worth to add a satellite photo in section 2.1. A map could probably help to understand in which pattern the fertilizer was applied in section 2.2.

3) Was it considered to measure the actual plant growth during the experiment period with eg LIDAR and create a point cloud of each of the plants? Perhaps drone mapping to consider in future work, potentially supported by multispectral imaging and creating a database/individual profile for each of the plants for the experimental period, as the plants are separated enough from each other that by using RTK a dedicated plant profile could be assigned.

4) Is it correct to conduct that for the 4 developmental stages, 4 different fertilizers should be used with different NPK ratios?

5) How was photosynthetic duration set on 8 hours a day? Was it considered as average during the whole vegetation period or it was somehow measured?

6) In section 2.3, again it would be reasonable to use a satellite photo to describe how and from where were the soil samples taken from.

7) Excellent that for the subsections in results section you have comprehensive description of each, followed by figures and tables.

8) As for the final results, maybe it would be worth to consider an extra subsection which would summarize a proper fertilization strategy that covers the whole vegetation period including all 4 growth stages. This would significantly increase the impact and contribution of this paper.

9) Based on the discussion, the authors have received similar results with other researchers, indicating that there are indeed patterns, such as conventional NPK ratio, which prioritizes N through the vegetation period. Such matchings indicate, that the results provided by the authors are valid.

10) What kind of differences (if any) would it be to expected when using solid granulated fertilizer instead of liquid fertilizer?

11) As stoichiometry seems to be one of the keywords in the paper, perhaps it would be worth to explain further in the introduction, what is this stoichiometric fertilization.

12) The list of references is time relevant, of which many sources are from the last 5 years. The sources have been chosen also to mainly match geographical location.

**Do you want your identity to be public for this peer review?** For information about this choice, including consent withdrawal, please see our Privacy Policy

Reviewer #1: No

Reviewer #2: No

---

## [Author Response · Author response to Decision Letter 1]

20 Sep 2025

Dear Editors and Reviewers,Thank you for your review of this manuscript and for your valuable amendments. We have carefully considered all the feedback and made the necessary revisions to the manuscript accordingly. Below, you will find point-by-point responses to each comment.

Reviewer #1:

1�In the Materials and Methods section, please specify the instruments and tools used to measure photosynthetic rate and leaf area. Details such as brand, model, and measurement protocols are important for reproducibility.

Thanks to the teacher 's suggestions, I have supplemented the relevant content in the material and method section.

2�In the Experimental Design subsection, although you provided the critical nutrient concentrations in leaves, you did not clearly explain how these values were used to calculate the theoretical nutrient requirements of the plants. Additionally, please justify why nutrient concentrations in leaves alone were considered, rather than including other plant organs (e.g., fruits, roots, stems). A whole-plant nutrient accounting approach would provide a more comprehensive estimation of actual requirements.

Thank you for the questions raised by the teacher regarding the calculation of leaf nutrient data. I have provided the relevant instructions below Table 1. Additionally, it is important to address why this experiment focuses solely on leaves and not other plant organs. Leaves serve as the primary organs for photosynthesis, transpiration, respiration, and other physiological metabolic processes in plants. To meet the growth requirements of plants, a significant amount of nutrients is transferred or transported to the leaves, while stems, branches, and roots primarily function in nutrient transport and absorption, storing only a minimal amount of nutrients. Consequently, the nutrient content and stoichiometric ratio of plant leaves most accurately reflect their overall nutritional status, growth rate, and nutrient utilization and limitations.

3�Furthermore, clarify how the fertilizer application rates (in Table 3) were calculated from the theoretical NPK nutrient demands. The conversion from nutrient requirement to actual fertilizer quantity should be detailed, including molecular weights, purity, and fertilizer formulations used.

Thanks to the teacher's inquiry, although the calculation formula was provided in the experimental design, the complexity of the calculations necessitates a clearer illustration. Therefore, I will present an example to elucidate the data calculation process. Below is the calculation process for the amount of N fertilizer applied during the first fertilization of treatment 1F, specifically for S1, over a 10-day period. Due to the multiple iterations involved in the calculation process, the resulting values exhibit slight deviations from those presented in the table.

4�Check for missing or improperly expressed units throughout the text and in all tables. Consistency and clarity in unit expression are essential for interpreting results correctly.

Thanks to the teacher 's advice, I have checked and corrected the full text.

5�Clarify the fertilizer treatment levels. Does "1F" represent the exact theoretical nutrient demand (i.e., the minimum requirement for optimal mineral nutrition of the crop)? Please explain the rationale behind the 1F, 5F, 10F, and 15F treatments.

Thanks to the teacher's question, 1F represents the minimum demand for optimal mineral nutrition in crops. Due to the uncertainty regarding the absorption and utilization efficiency of fertilizers by blueberry plants, as well as the effective content conversion efficiency of fertilizers applied to the soil, discrepancies arise between the theoretical and actual fertilizer requirements. Consequently, it is essential to conduct various gradient fertilization tests to determine the precise amount of fertilizer needed.

6�In the Discussion on nutrient uptake, carefully re-evaluate the conclusion that nitrogen (N) is the most important element during the fruit expansion stage. In many fruit crops, nitrogen primarily supports vegetative growth and vigor, while potassium (K) is more closely associated with fruit size, firmness, and flavor, and phosphorus (P) with root development and flowering. Consider whether your findings for blueberry differ from this general pattern, and if so, explain the physiological basis.

Thank you for your insightful question. Given that the ratios of nitrogen, phosphorus, and potassium in this experiment are fixed, and the article does not provide any data related to the fruit, discussing fruit-related content in the discussion section may not hold significant relevance.

7�Strengthen the Discussion section further by adding more context, critical analysis, and possible future research perspectives.

Thanks for the advice given by the teacher, I have made corresponding adjustments in the discussion section.

8�Please correct the many syntax, grammar, and vocabulary errors present in the manuscript. The current language quality significantly affects the clarity and readability of the paper and should be revised by a fluent English speaker or professional editor.

I appreciate the teacher's feedback regarding the grammar, vocabulary errors, and other issues in the article. I have made the necessary revisions accordingly.

Reviewer #2:

1�As the it was described, the 7 year old plants of "Powderblue" cultivar are up to 85 cm tall. If and how could be the research results interpreted in order to apply NPK for smaller cultivars, such as Vaccinium angustifolium?

The amount of fertilizer applied in this experiment was primarily determined by the size of the plant leaves and their photosynthetic rates. To formulate fertilization strategies for other plants based on the results of this study, one can calculate the appropriate amounts according to their respective leaf areas and photosynthetic rates.

2�Which was the plantation pattern or the layout? What was the distance from the plants in row and column? Perhaps worth to add a satellite photo in section 2.1. A map could probably help to understand in which pattern the fertilizer was applied in section 2.2.

Thank you for the questions raised by the instructor regarding the layout of the blueberry test plants. Although no photographs were taken during the experiment, I have provided a detailed textual description in the experimental design section.

3�Was it considered to measure the actual plant growth during the experiment period with eg LIDAR and create a point cloud of each of the plants? Perhaps drone mapping to consider in future work, potentially supported by multispectral imaging and creating a database/individual profile for each of the plants for the experimental period, as the plants are separated enough from each other that by using RTK a dedicated plant profile could be assigned.

Thanks to the teacher's suggestion, the issues previously overlooked in the experiment have been addressed. I believe this recommendation is highly insightful and will significantly benefit my future experiments. If circumstances allow, I will consider incorporating this content into my subsequent research.

4�Is it correct to conduct that for the 4 developmental stages, 4 different fertilizers should be used with different NPK ratios?

Thanks to the teacher's inquiry, it is important to note that the ratio of carbon, nitrogen, phosphorus, and potassium is fixed in this experiment. Although the amount of fertilizer applied varies at different stages, the ratio of nitrogen, phosphorus, and potassium remains consistent throughout.

5�How was photosynthetic duration set on 8 hours a day? Was it considered as average during the whole vegetation period or it was somehow measured?

In response to the teacher's inquiry, the reason for calculating the photosynthetic duration at 8 hours in blueberry cultivation and research primarily aligns with the physiological characteristics of blueberries, which prefer sunlight. The light saturation point for blueberries is relatively low; thus, the intense midday light not only becomes unusable but can also inhibit photosynthesis and even damage the leaves. Consequently, these 8 hours represent the 'core period' of the day—specifically the morning and afternoon—when the light is softer and photosynthesis is most efficient. Therefore, this study adopts an 8-hour calculation for photosynthesis.

6�In section 2.3, again it would be reasonable to use a satellite photo to describe how and from where were the soil samples taken from.

Thanks to the suggestions provided by my instructor, I have made corresponding modifications to the sample collection and determination section, and I have included a diagram illustrating the sample collection process.

7�Excellent that for the subsections in results section you have comprehensive description of each, followed by figures and tables.

Thanks to the teacher 's recognition, I will make more efforts to improve the content of the article.

8�As for the final results, maybe it would be worth to consider an extra subsection which would summarize a proper fertilization strategy that covers the whole vegetation period including all 4 growth stages. This would significantly increase the impact and contribution of this paper.

Thank you for the teacher's suggestion; it is indeed valuable. However, due to the insufficient amount of data presented in the article, it is challenging to convincingly support the proposed fertilization strategy.

9�Based on the discussion, the authors have received similar results with other researchers, indicating that there are indeed patterns, such as conventional NPK ratio, which prioritizes N through the vegetation period. Such matchings indicate, that the results provided by the authors are valid.

I sincerely appreciate the recognition from the reviewer regarding the validity of this study's results. The findings related to the absorption of nitrogen, phosphorus, and potassium align with previous research, thereby enhancing the reliability and persuasiveness of our conclusions.

10�What kind of differences (if any) would it be to expected when using solid granulated fertilizer instead of liquid fertilizer?

In response to the teacher's inquiry, this experiment utilized high-purity chemicals rather than conventional fertilizers. Should you wish to incorporate other fertilizers, adjustments to the quantity can be made based on the varying concentrations of nitrogen, phosphorus, and potassium present in the selected fertilizers.

11�As stoichiometry seems to be one of the keywords in the paper, perhaps it would be worth to explain further in the introduction, what is this stoichiometric fertilization.

Thanks to the teacher 's suggestions, I have added relevant content to the introduction of the article.

12�The list of references is time relevant, of which many sources are from the last 5 years. The sources have been chosen also to mainly match geographical location.

We appreciate the recognition from our instructor. In terms of literature selection, we prioritize research published within the last five years. Furthermore, we focus on literature pertinent to the regional environment to enhance the applicability and reference value of our research findings. We will continue to adhere to this method of literature selection.

---

## [Decision Letter · Decision Letter 1]

8 Oct 2025

Response of Soil Nutrient Dynamics and Stoichiometric Characteristics in Blueberry to Fertilization Rates

PONE-D-25-17493R1

Dear Dr. Wang,

We’re pleased to inform you that your manuscript has been judged scientifically suitable for publication and will be formally accepted for publication once it meets all outstanding technical requirements.

Kind regards,

Rachid Bouharroud

Academic Editor

PLOS ONE

Additional Editor Comments (optional):

Reviewers' comments:

Reviewer's Responses to Questions

**Comments to the Author**

Reviewer #1: All comments have been addressed

Reviewer #2: All comments have been addressed

2. Is the manuscript technically sound, and do the data support the conclusions?

Reviewer #1: (No Response)

Reviewer #2: Yes

3. Has the statistical analysis been performed appropriately and rigorously?

Reviewer #1: Yes

Reviewer #2: Yes

4. Have the authors made all data underlying the findings in their manuscript fully available?

Reviewer #1: Yes

Reviewer #2: Yes

5. Is the manuscript presented in an intelligible fashion and written in standard English?

Reviewer #1: Yes

Reviewer #2: Yes

Reviewer #1: (No Response)

Reviewer #2: Thanks for the authors for the revisions and comments. Everything has been addressed. The authors revisions enhanced the quality of the paper significantly. Of course, everything could be always enhanced and fine-refined, but in this case I think the further gains would be marginal. Good luck for the authors in their future research (and fingers crossed that you will perform a similar comprehensive research on lowbush blueberry plants).

**Do you want your identity to be public for this peer review?** For information about this choice, including consent withdrawal, please see our Privacy Policy

Reviewer #1: No

Reviewer #2: No

---

## [Editor Report · Acceptance letter]

PONE-D-25-17493R1

PLOS ONE

Dear Dr. Wang,

I'm pleased to inform you that your manuscript has been deemed suitable for publication in PLOS ONE. Congratulations! Your manuscript is now being handed over to our production team.

Kind regards,

on behalf of

Dr. Rachid Bouharroud

Academic Editor

PLOS ONE